# Experimental Evaluation of Hoop Stress–Strain State of 3D-Printed Pipe Ring Tensile Specimens

Milan Travica [1,*], Nenad Mitrovic [2], Aleksandar Petrovic [2], Isaak Trajkovic [1], Milos Milosevic [1], Aleksandar Sedmak [2] and Filippo Berto [3]

1   Innovation Center of the Faculty of Mechanical Engineering, 11000 Belgrade, Serbia
2   Faculty of Mechanical Engineering, University of Belgrade, 11000 Belgrade, Serbia
3   Faculty of Engineering, Norges Teknisk-Naturvitenskapelige Universitet (NTNU), 7070 Trondheim, Norway
*   Correspondence: mtravica@mas.bg.ac.rs

**Abstract:** Data on the strain and stress status of the pipe in the circumferential direction are required for various pipe manufacturing procedures (e.g., in the oil business, the process of manufacturing seamless pipes with a conical shaft). The aim of this study is to develop a procedure to determine the strain and stress behavior of Pipe Ring Tensile Specimens (PRTSs) in the hoop direction, as there are a lack of official standardized methods for testing PRTS. This paper discusses the application of the Digital Image Correlation method for testing plastic PRTSs. PRTSs are tested using a specially designed steel tool with two D blocks. A 3D-printed PRTS is placed over two D-shaped mandrels, which are fixed on a tensile tool and tensile testing machine. The strain evolution in the gage length of the specimens is captured using the three-dimensional Digital Image Correlation (3D DIC) method. To check the geometry of the cross-section of the PRTS after fracture, all the specimens are 3D scanned. For the study, six groups of PRTS are analyzed, consisting of three filling percentages (60, 90, and 100%) and two geometry types (Single and Double PRTS). The results show that the type and percentage of filling, as well as the method of printing, affect the material behavior. However, the approach with the DIC system, 3D printer, and scanner shows that they are effective instruments for mapping complete strain fields in PRTS, and thus are effective in characterizing the mechanical properties of pipes.

**Keywords:** pipe ring tensile specimen; digital image correlation method; 3D printing; tensile test; strain; tensile testing tool; 3D scanning

## 1. Introduction

Different pipe production processes need data regarding the stress state of the pipe in the circumferential direction. The first example of such a process is the process of forming seamless pipes using a conical shaft in the oil industry. This is a process whereby a conical shaft is inserted into the formed glowing sample, which, by passing through the sample, radially expands the pipe to the desired diameter. The second is the production of polymer pipes, which causes a significant difference in the mechanical behavior between traditional tensile test specimens from basic materials and the final product of plastic pipes [1]. Due to the long distance and complex terrain of pipeline transportation, high toughness, fatigue resistance, corrosion resistance, and fracture resistance are necessary properties of pipeline materials [2].

Standard procedures for measuring material tensile properties are performed according to ASTM A370 [3], ASTM E8 [4], and ASTM D638-14 [5]. Most often, the specimens are cut from a plate or a pipe. In the case of pipes, there is an additional preparation process, i.e., flattening the specimen to be placed on the tensile testing machine. Studies [6,7] in which the effects of the flattening process have been analyzed prove that it significantly affects the material properties. Flattening causes pre-strains and residual stresses in the

material, which influence the material behavior of pipes and, as a result, the accuracy of material parameter identification. To address these issues, some researchers devised the Ring Hoop Tensile Test (RHTT), which was originally designed to determine the hoop mechanical properties of thermoplastic tubular materials [8].

Price [9] was the first to propose the RHTT technique in 1972 to explore the effect of hydride precipitation in internally pressured pipes. Using a tensile testing machine, Arsene and Bai (1996, 1998) [10] presented a RHTT to properly assess the hoop properties of a pipe. RHTT is a procedure for testing pipe rings using tools with D blocks [9]. A dog bone specimen geometry is machined on a ring taken from the pipe and placed over two D-shaped mandrels that are parted using a tension testing machine. In Ref. [11], the geometry of the test specimen was optimized using FEM. The optimized pipe specimen according to [11] has a better load distribution and crack and fracture development in the center measuring region. The fundamental benefit of the RHTT approach is that it can be used to examine the pipe material properties of as-received metallic pipes along the hoop direction, eliminating pre-strains and residual stresses by flattening.

The process of cutting a standard (dog bone) specimen requires enough material on the pipe, which is problematic for pipe dimensions smaller than DN100, and for the pipes, there is an additional preparation process that influences the real material's mechanical properties. One paper [6] discusses the application of the Digital Image Correlation (DIC) method for testing Pipe Ring Tensile Specimens (PRTS). PRTS are tested using a specially defined PRTS testing tool with two D blocks. With some tools, proposed in other works [6,11,12], there is a problem with specimen rotation and the bending effect.

There are several available studies [12–14] that use the DIC method in the process of RHTT. DIC is an optical, non-contact method that can provide the full 3D strain/displacement field [15,16]. DIC is independent of the material tested, as well as of the shape of the object, which enables its application in various fields, such as structural integrity, biomaterials, and polymers [17–27]. It is important to note that in all the studies mentioned before, one camera was used for the displacement and strain analysis. The use of a single camera for the DIC method is possible for in-plane test cases. However, the geometry of the pipe is cylindrical, and therefore the displacement and strain analysis with one camera on such a surface is questionable.

The aim of this study is to develop specimen geometry, testing tools, and a procedure to determine the strain and stress behavior of PRTS in the hoop direction using DIC and a 3D scanner.

## 2. Materials and Methods

In this study, a 3D printer (German RepRap GmbH, Feldkirchen, Germany) was used to produce the PRTS, as additive manufacturing is currently one of the promising methods for the fabrication of products of complex shapes [28]. The reason for using a 3D printer was the fast manufacturing of all PRTSs. The specimens were printed from polylactic acid (PLA) plastic material on a RepRap x400 3D printer (German RepRap GmbH, Feldkirchen, Germany) to develop a procedure for testing PRTS. PLA is an available bio-based polymer [29] that is commonly used for Fused Deposition Modeling. It has good dimensional accuracy and is suitable for rapid specimen fabrication.

The dimensions of the PRTS were taken for the pipe DN32 (Ø42.4 mm × 2.8 mm), and the dimensions and appearance of the standard specimen were taken from the ASTM A370 standard [3]. Two types of test specimens were made for the procedure development:

1. Single PRTS with standard specimen configuration on one side (Figure 1a–d);
2. Double PRTS with standard specimen configuration on two sides with a mutual angle at 180° (Figure 1e–h).

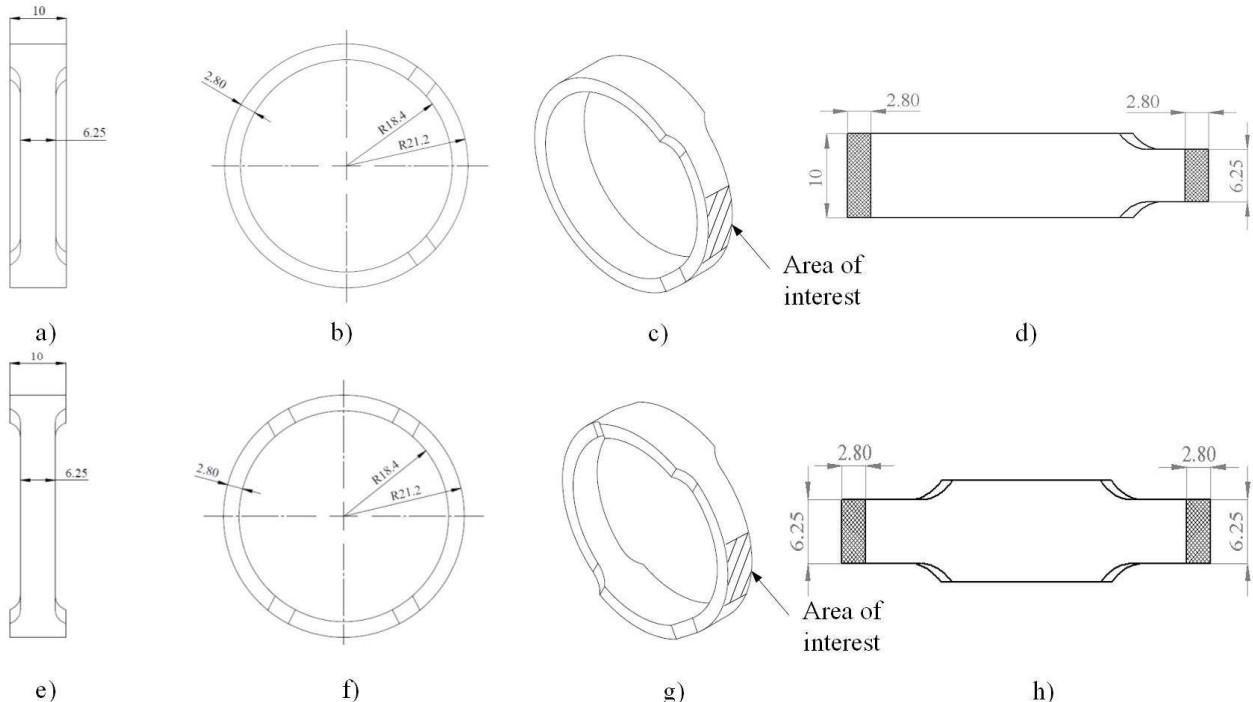

**Figure 1.** Pipe Ring Tensile Specimen dimensions DN32 (Ø42.4 × 2.8 mm): (**a**) front view Single PRTS; (**b**) side view Single PRTS; (**c**) isometric view Single PRTS; (**d**) nominal cross-section area Single PRTS; (**e**) front view Double PRTS; (**f**) side view Double PRTS; (**g**) Isometric view Double PRTS; (**h**) nominal cross-section area Double PRTS.

In total, 30 PRTS were prepared, five per each of six PRTS groups in three different filling percentages (60, 90, and 100%) and with two geometry types (Single and Double PRTS). All the specimens were printed with a honeycomb structure. For the DIC method, all PRTS were painted with white paint as a base color for spraying a stochastic pattern of black dots prior to the experiment.

The hoop tensile strength of a composite pipe specimen was measured experimentally using the test method with D blocks [30]. A PRTS testing tool with D blocks was created for the testing and development of the PRTS testing procedure. The PRTS testing tool was made of X20CrMoV12-1 steel, as shown in Figure 2 (position 2). It consisted of two forks (Figure 2, positions 4 and 5), which were inserted into the tensile testing machine. Two D blocks (Figure 2, position 6) were inserted into the PRTS (Figure 2, position 7, and detail A) and assembled on the tool, as illustrated in Figure 2. The D blocks simulated the internal pressure in the PRTS, which is the most common load on the pipeline. The PRTS testing tool used a fork to transfer the load from the tensile testing machine (Figure 2, position 1) to the D blocks and PRTS.

PRTS were tensile tested in a Shimadzu Autograph AGS-X Series tensile testing machine (Shimadzu, Kyoto, Japan) with maximum test loads of 100 kN. The tensile testing machine was set for testing according to [3] with a test speed of 1 mm/min for all PRTS. During the experiment, stroke displacements and force values were monitored using the Trapezium software. Tensile test in order to determine the apparent hoop tensile strength [31].

Strain fields are measured during the specimen using stereo Digital Image Correlation (DIC) system [32]. The Aramis 2M system (GOM, Braunschweig, Germany) was used to conduct strain field experiments. The Aramis 2M is based on the three-dimensional Digital Image Correlation (3D DIC) method. Calibration was conducted as part of the pre-experimental preparations. Several studies [21–27] discuss the detailed mode of operation, calibration, preparation of the measuring object, and measuring technique. The Aramis setup parameters for this study were as follows:

- Two CCD cameras with a resolution of 1600 × 1200 pixels;

- Two 50 mm camera lenses;
- Measuring volume of 105 mm $\times$ 75 mm $\times$ 55 mm;
- Measuring distance (distance between camera support and center of measuring volume) of 800 mm;
- Facet (subset) size of 25 $\times$ 20 pixels;
- Calibration point variability of 0.031 pixels (for correct calibration, the manufacturer states that calibration deviation may be between 0.01 and 0.04 pixels);
- LED lamp for specimen lighting.

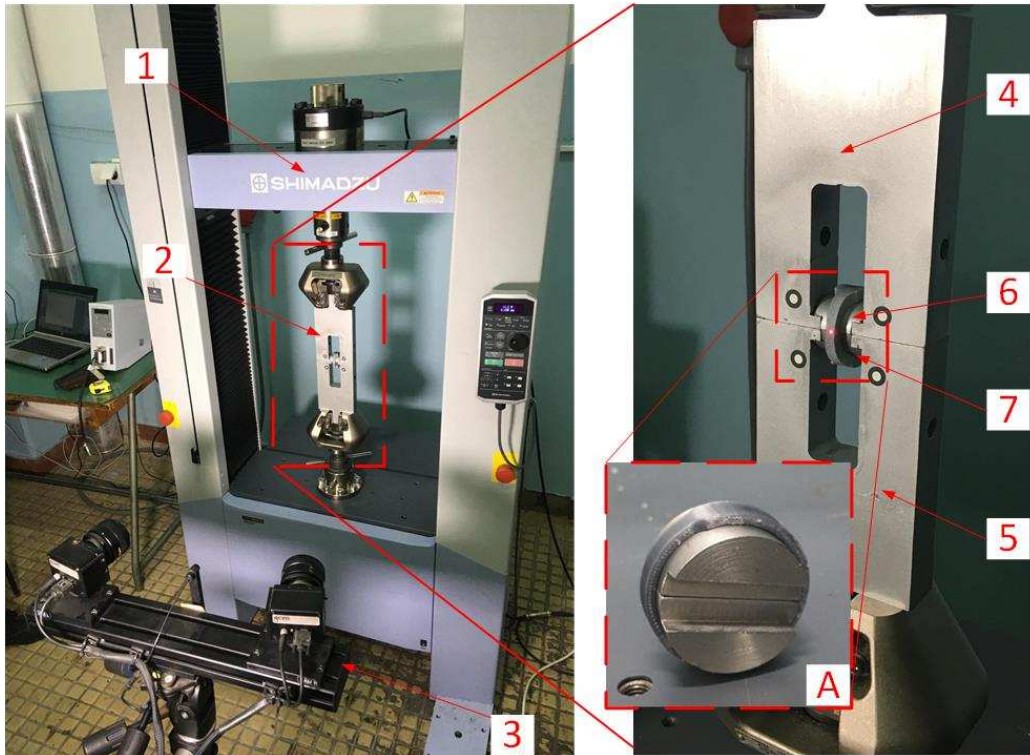

**Figure 2.** Detailed view of testing setup: (1) tensile testing machine; (2) PRTS testing tool; (3) Aramis 2M; (4) upper fork; (5) lower fork; (6) two D- blocks; (7) PRTS, label (A) assembly of PRTS and D blocks.

To check the geometry of the cross-section of the PRTS after the tensile test, a 3D scanner Atos Core 200 (GOM, Braunschweig, Germany) was used. The Atos Core 200 setup parameters were as follows:

- Measuring surface of 200 $\times$ 150 mm;
- Working distance of 250 mm;
- Sensor dimensions of 206 $\times$ 205 $\times$ 64 mm.
- The test procedure consisted of the following steps:
- Specimen preparation. All PRTSs were painted with white paint as a base color and a stochastic pattern of black dots, as the measuring surface must have a high contrast pattern to clearly allocate the pixels in images.
- DIC system calibration, including adjusting and calibrating the cameras.
- Tool and specimen positioning, including placing the testing tool on the tensile testing machine, assembling the D blocks inside the PRTS, and mounting the assembly on the testing tool.
- Tensile testing machine setup, including defining the test parameters on the tensile testing machine.
- DIC measurement—after successful calibration, a DIC measurement was carried out. The tensile test procedure was conducted according to the definition outlined in

Standard [3]. The testing installation scheme is illustrated in Figure 2. Digital images were recorded manually immediately every 1 s during the loading. The first recorded image (before the loading) was the nominal image for data processing.

- DIC data processing—afterwards, computation was performed using the Aramis software.
- Three-dimensional scanner calibration, including adjusting and calibrating the cameras.
- Three-dimensional scanner measurement. After successful calibration, a measurement was carried out.
- Three-dimensional data processing. Afterwards, computation was performed using the Atos software (GOM, Braunschweig, Germany).

## 3. Results and Discussion

The fracture of all single PRTSs occurred on the side on which the camera was recording. In the case of Double PRTS, the fracture place occurred on both sides. The diagram in Figure 3 shows the average value of break force for all PRTSs with standard deviation. Both Single and Double PRTS showed negligible differences between 90 and 100% infills. However, there was a significant difference for both PRTS types with 60% infill compared to with 90 and 100% infill. The recommendation for future experimental studies is to use the 90% infill specimens, as their break force is similar to the 100% infill case and their time for printing, as well as material consumption, is reduced.

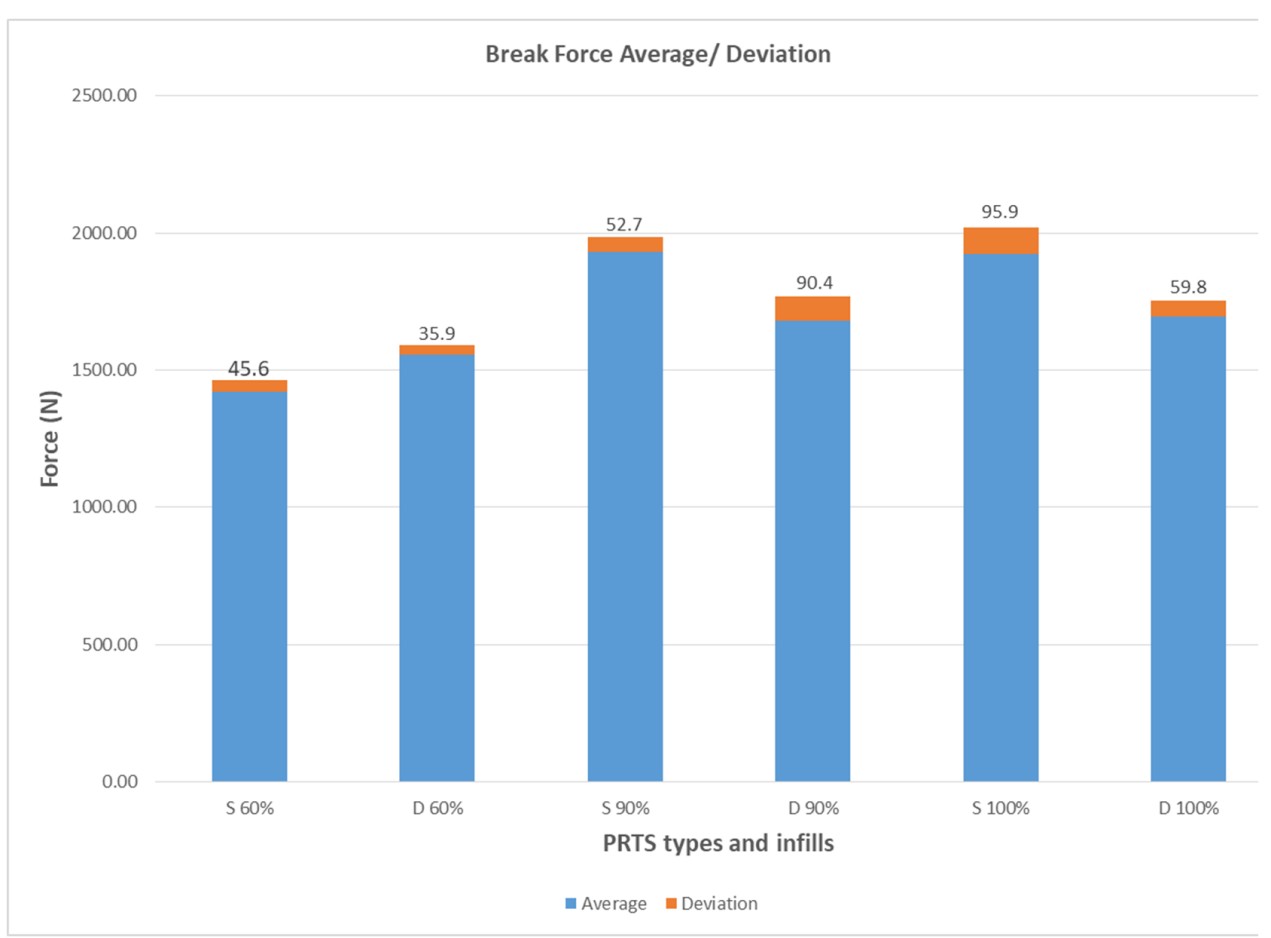

**Figure 3.** Average and standard deviation of the break force for all infills of Pipe Ring Tensile Specimens.

The PRTS cross-sectional dimensions after fracture were analyzed using the Atos Core 200 (GOM, Braunschweig, Germany). As illustrated in Figure 4, the dimensions were analyzed on both sides of the PRTS. Each side was measured in two orthogonal directions—one direction in specimen width (lines A1–A2 and A3–A4) and one direction in specimen thickness (lines B1–B2 and B3–B4). Lines A1–A2 and B1–B2 were positioned

on the specimen side where the area of interest was analyzed. All measuring lines were placed at specimen edges. Figure 4 shows a Single PRTS, with 90% infill, showing the measurement points and analyzed distances.

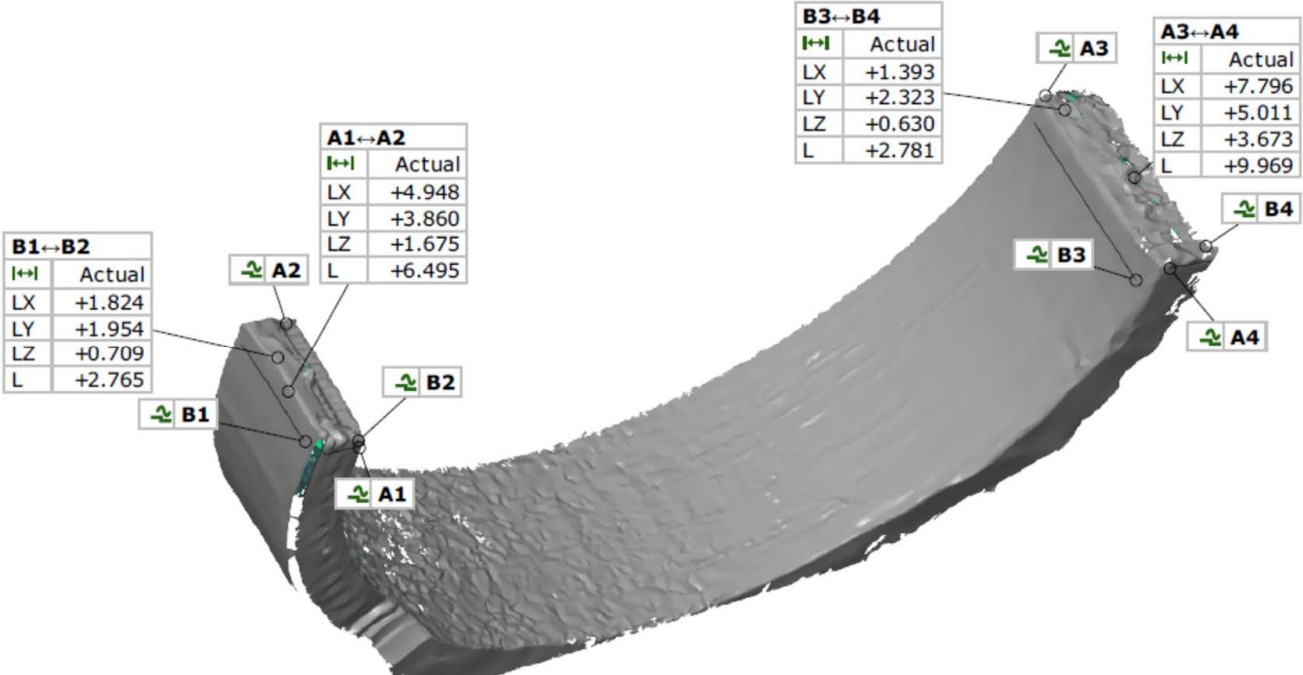

**Figure 4.** Atos Core 3D scan of Single PRTS, filled to 90% with specified measuring elements and cross-sectional dimensions.

Table 1 shows the percentage changes in cross-sectional dimensions for all types of PRTS. The results of the change in cross-sectional dimensions indicated a negligible difference between PRTS with 90% and 100% filling. The differences were more noticeable for PRTS with 60% infill compared to 90 and 100% PRTS infill.

**Table 1.** PRTS dimension changes in percentages.

| Single PRTS 60% | | | | Double PRTS 60% | | | |
|---|---|---|---|---|---|---|---|
| A1–A2 | A3–A4 | B1–B2 | B3–B4 | A1–A2 | A3–A4 | B1–B2 | B3–B4 |
| +3.32% | −0.3% | −1.27% | −0.6% | +0.14% | +0.079% | −1.17% | +1.37% |
| Single PRTS 90% | | | | Double PRTS 90% | | | |
| A1–A2 | A3–A4 | B1–B2 | B3–B4 | A1–A2 | A3–A4 | B1–B2 | B3–B4 |
| +3.83% | −5.02% | −9.05% | −5.02% | +2.53% | +2.12% | −3.32% | −4.98% |
| Single PRTS 100% | | | | Double PRTS 100% | | | |
| A1–A2 | A3–A4 | B1–B2 | B3–B4 | A1–A2 | A3–A4 | B1–B2 | B3–B4 |
| +3.74% | −4.98% | −8.85% | −5.33% | +3.03% | +1.07% | −3.55% | −5.12% |

For all PRTS types and filling percentages, the engineering stress was calculated using the force data from the tensile testing machine for all stages and nominal cross-section areas. The strain was obtained using the Aramis 2M system. The engineering stress–strain diagrams are presented in Figure 5a,b. Figure 5a shows a stress–strain diagram for nominal specimen cross-section dimensions without the percentage infill impacts. To consider the effect of filling on the stress value, Figure 5b shows a stress–strain diagram with corrected cross-section areas and correction factors of 0.6, 0.9, and 1.0 for PRTSs with 60%, 90%, and

100% infill, respectively. All stress values are in MPa and strain is given in percentages. It can be seen from the diagram in Figure 5a,b that the fracture occurred after reaching the maximum stress value, which indicated that it was a distinctly brittle fracture. The stress values for Single PRTS for all filling percentages were lower than the stress values of Double PRTS, as the loaded cross-sectional area for Double PRTS was smaller than that of Single PRTS. The stress–strain curve differences and similarities were more noticeable when presented with correction factors.

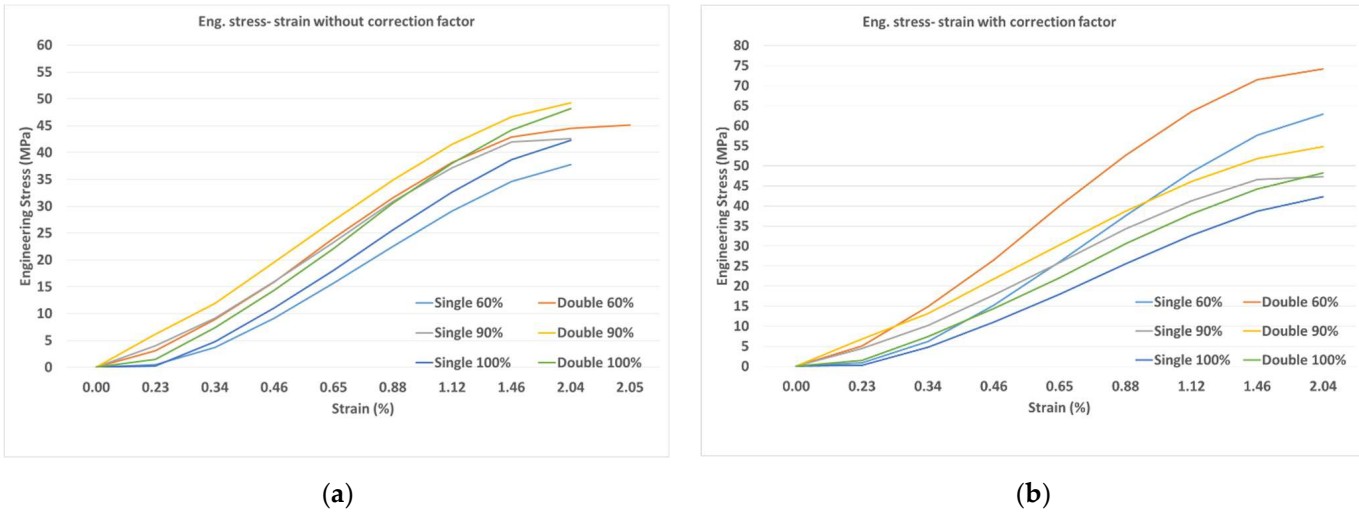

(**a**)                       (**b**)

**Figure 5.** Engineering stress–strain: (**a**) without correction factor; (**b**) with correction factor of 0.6 for PRTS 60% infill, 0.9 for PRTS 90% infill, and 1.0 for PRTS 100% infill.

The results of the von Mises strain are presented in this study for the DIC measurement. The strain field (Figures 6a and 7a) was analyzed with two sections (Sections 0 and 1) and four points (stage points 0–3). Section 0 was placed horizontally (black line) and was located in the junction of the D blocks. Section 1 was orthogonal to Section 0, placed vertically on the area of interest (yellow line). Stage points were placed at the ends of all sections. Stage points 0 and 1 were placed on Section 0 and stage points 2 and 3 were placed on Section 1. Figures 6a and 7a show a visualization of the von Mises strain field just before the fracture for Single and Double PRTS with 90% infill as a function of section length, respectively. The experimental results presented in Figures 6b and 7b show that the highest strain values were registered at the area of interest in the D block junction. The highest von Mises values were 11.2% (Single PRTS 90% infill) and 9.4% (Double PRTS 90% infill), which are shown as the peak (yellow line) in Figures 6b and 7b. The diagrams in Figures 6c and 7c show the von Mises strain as a function of time for four stage points. The stage point diagram (Figure 6c) has a similar trend, which is constant from the beginning to the thirtieth stage, but the strain values exponentially increase from the thirtieth stage to the fracture with a maximum von Mises strain of 2,4% (Single PRTS 90% infill). The stage point diagram (Figure 7c) shows a similar trend for stage points 0–1 and 2–3, which are approximately linear from the beginning to the fracture, with a maximum von Mises strain of 2% (stage points 2–3), and constant from the beginning to the sixtieth stage, but the strain values exponentially increase from the sixtieth stage to the fracture with a maximum von Mises strain of 5.76% (stage points 0–1).

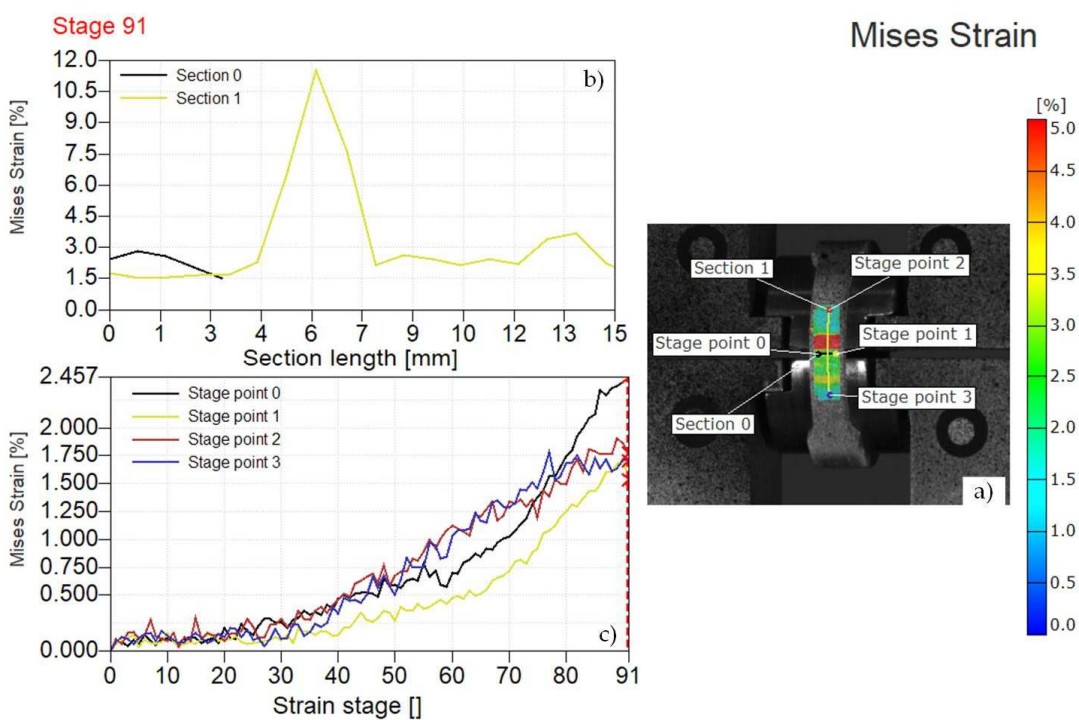

**Figure 6.** Visualization of von Mises strain field before fracture for Single PRTS specimen with 90% infill: (**a**) visualization of von Mises strain field at the stage just before fracture; (**b**) von Mises strain value with respect to the section length; (**c**) von Mises strain value change over time.

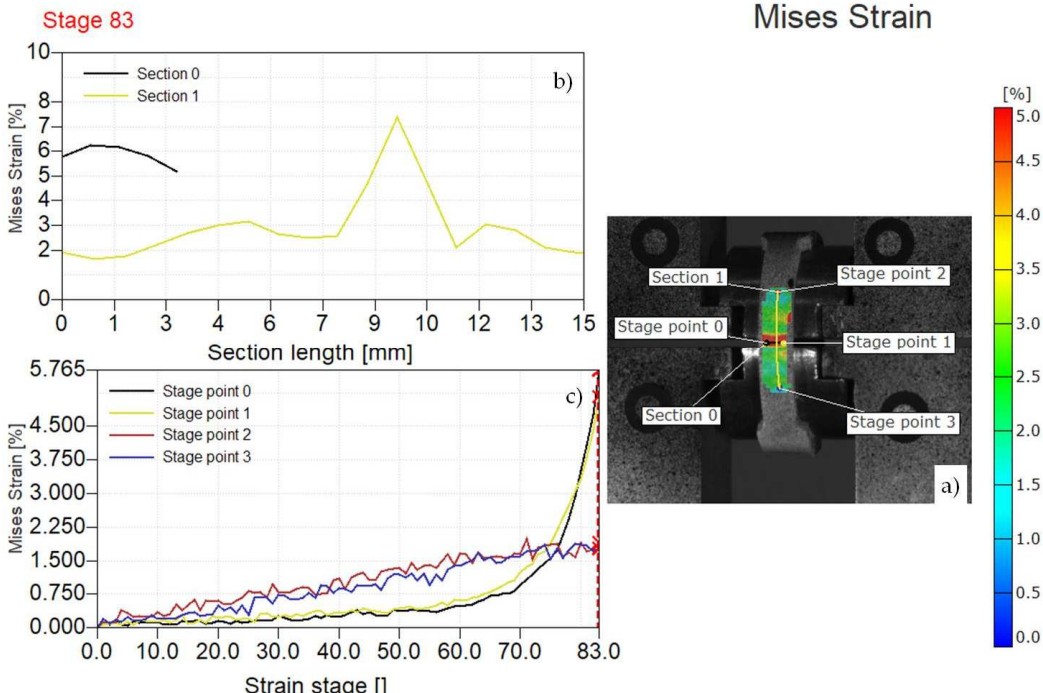

**Figure 7.** Visualization of von Mises strain field before fracture for Double PRTS specimen with 90% infill: (**a**) visualization of von Mises strain field at stage just before fracture; (**b**) von Mises strain value with respect to the section length; (**c**) von Mises strain value change over time.

The diagrams in Figure 8a,b show average strain-stage dependence with standard deviation values for PRTS with 90% and 100% infill. The values for Single and Double PRTS with 60% infill are not graphically represented due to high result variations, as shown in previous paragraphs. The diagram in Figure 8a shows an average strain-stage dependence

with standard deviation values at stage points 0–3 for Single PRTS with 90% and 100% infill. The graph in Figure 8a shows a similar trend of von Mises strain and standard deviation values for Single PRTS with 90% and 100% infill, which have an approximately linear trend. The diagram in Figure 8b shows average strain-stage dependence with standard deviation values at stage points 0–3 for Double PRTS with 90% and 100% infill. The graph in Figure 8b shows similar a trend of von Mises strain and standard deviation values for Double PRTS with 90% and 100% infill, which have an approximately linear trend. The difference in standard deviation value for Single PRTS with 90% and 100% infill (Figure 8a,b) is observed before fracture (from the seventy-fifth stage to the end), where the Single and Double PRTS with 90% infill have slightly higher values (average 38.96%) than Single and Double PRTS with 100% infill.

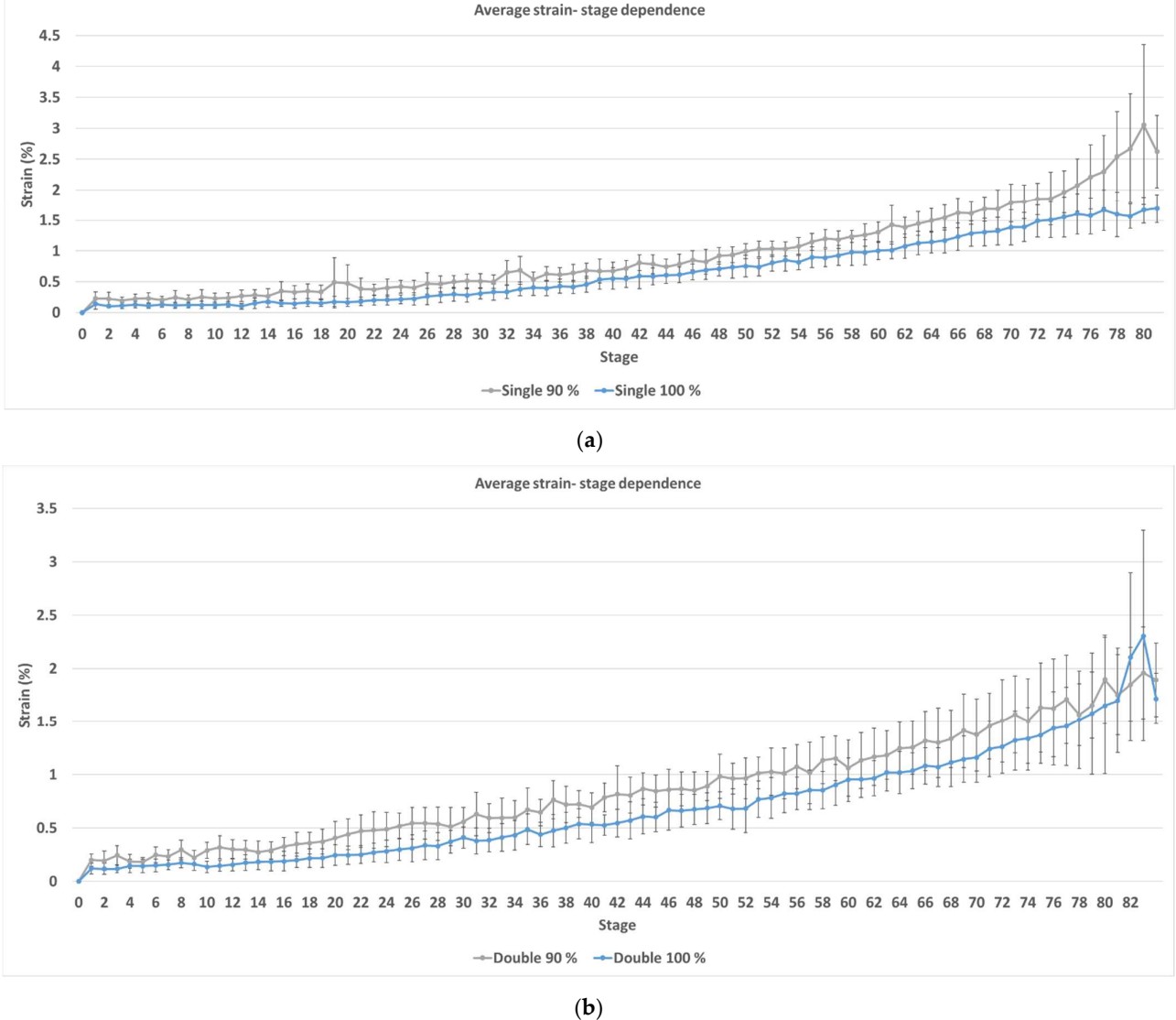

**Figure 8.** Visualization of average strain-stage dependence for Single and Double PRTS with 90% and 100% infill with standard deviation: (**a**) Single PRTS 90% and 100% infill; (**b**) Double PRTS 90% and 100%.

Diagram 8a shows that the deformation value for Single PRTS with 90% infill is higher than that for Single PRTS with 100% infill on average by 0.34% for each stage, but that Single PRTS with 100% infill has a more linear strain-stage dependence than Single PRTS with 90% infill. Diagram 8b shows that the deformation value for Single PRTS with 90% infill is higher than that for Single PRTS with 100% infill on average by 0.17% for each stage,

but that Single PRTS with 100% infill has a more linear strain-stage dependence than Single PRTS with 90% infill.

The limitations of this study are the PRTS material used, the infill percentage of PRTS material, the location of the fracture, and equipment limitations. The behavior of PLA plastic does not reflect the characteristics of conventional pipe materials because of the pronounced occurrence of brittle fracture, but it is sufficient to develop the procedure that will later be applied to the steel PRTS. The results of stress–strain and cross-sectional dimensions indicate a negligible difference between PRTS with a 90% and 100% filling. The differences are more noticeable for PRTS with 60% infill compared to that with 90 and 100% PRTS infill. The application of DIC cameras in Double PRTS is questionable because the location of the fracture is not predetermined. The application of the DIC method is recommended for Single PRTS. The 3D DIC method has some limitations. Correct 3D calculation and strain computation for sample edges are not achievable since the 3D computation of the measuring points is based on pixels that need to be observed from the right and left cameras with the individual facet pattern. As a result, the high strain values (red color) seen on the strain field edges reflect system problems that are ignored.

## 4. Conclusions

The lack of standard methods for testing PRTS necessitates further development. From the presented experimental results, the following conclusions can be drawn:

- The DIC approach and the Aramis 3D optical system are effective instruments for mapping complete strain fields in PRTS, and thus are effective in characterizing pipe mechanical properties.
- The proposed methodology is shown to be applicable to Single PRTS.
- The results of the change in cross-sectional dimensions indicate a negligible difference between PRTS with a 90% and 100% filling. The differences are more noticeable for PRTS with 60% infill compared to that with 90 and 100% PRTS infill.
- Single PRTS 90% and 100% infill showed similar results, from which it can be concluded that for further research in the field of plastic PRTS, the recommendation is to work with Single PRTS 90% because the consumption of printing material is smaller and the printing time is shorter.

**Author Contributions:** Conceptualization, M.T. and N.M.; methodology, M.T. and N.M.; software, N.M.; validation, A.P., F.B. and A.S.; formal analysis, M.T.; investigation, M.T.; resources, I.T.; data curation, M.T.; writing—original draft preparation, M.T.; writing—review and editing, M.T.; visualization, M.T.; supervision, N.M. and M.M.; project administration, A.P.; funding acquisition, A.S. All authors have read and agreed to the published version of the manuscript.

**Funding:** This research was funded by [Ministry of Education, Science and Technological Development Republic of Serbia] grant number [451-03-68/2022-14/200105].

**Institutional Review Board Statement:** Not applicable.

**Informed Consent Statement:** Not applicable.

**Data Availability Statement:** Not applicable.

**Conflicts of Interest:** The authors declare no conflict of interest.

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
