# Peer review of "Experimental Evaluation of Hoop Stress–Strain State of 3D-Printed Pipe Ring Tensile Specimens"

_metals, doi:10.3390/met12101560_

Round 1

Reviewer 1 Report

Subjected article deals with development of the methodology for evaluation of the stress-strain properties in hoop orientation for  pipe ring tensile specimens.

The article is clearly written, however, there are several things that need to be addressed.

1. I understand the reasons for choosing of 3D printed PLA as an experimental material, but its behavior si far different to the standard materials for the pipes (this is also stated by the authors in lines 248-250) However, there is questionable, how the 3D DIC method will be able to reliable capture significantly larger deformations. Therefore, it would be valuable, if the authors could add some references showing this capability of the chosen system in the appropriate position in the text.

2. Figure 2 – for readers It may be helpful if the authors could add a schematic drawing of the testing configuration (D-blocks + PTRS)

3. Overall quality of graphs (Fig. 5, 6, 7,8 ) is poor.  Much more attention should be paid to the image preparation.

Fig.5  - the axis description has to be added. 

Fig.6 – annotation + scale in Fig.6a is not readable.

Fig.7 and 8 – axis description. Fig 7 and 8 show the same dependences, probably authors placed the wrong images, etc.

4. Lines 193-194: Correction factors were applied to normalize stress-strain data. The used approach is quite a simplification. Maybe it should be better to use correction factors based on the elastic modulus comparison.

5. Fig.5b and 6b – very steep changes in the graphs showing von Mises strain vs. Section length suggest, that the chosen resolution could not be sufficient to reveal real strain distribution within a given section. Is this a limitation of the method, or this could be improved by adjusting the parameters?

6. Fig.8 shows average strain vs. stage dependence. Representation of the average strain vs stress would be much more informative.

Author Response

Thank you for detailed comments.

All of these comments were analyzed and the answers are shown below:

  1. I understand the reasons for choosing of 3D printed PLA as an experimental material, but its behavior so far different to the standard materials for the pipes (this is also stated by the authors in lines 248-250. However, there is questionable, how the 3D DIC method will be able to reliable capture significantly larger deformations. Therefore, it would be valuable, if the authors could add some references showing this capability of the chosen system in the appropriate position in the text.

The paper states (lines 82,83,84) that the material was chosen for the reason of faster production of the test specimens and for the purpose of developing the procedure.
Within the defined measuring volume of the DIC method, there are no restrictions regarding the deformation value. Line 72 lists the literature where DIC was used as well as the types of materials, and among other things, polymers in which the values of deformation are significantly higher than those in our experiment. The mentioned method is applicable in this case.

  1. Figure 2 – for readers It may be helpful if the authors could add a schematic drawing of the testing configuration (D-blocks + PTRS)

Comment accepted and image corrected in the paper.

  1. Overall quality of graphs (Fig. 5, 6, 7, 8) is poor. Much more attention should be paid to the image preparation.

Comment accepted and image corrected in the paper.

Fig.5 - the axis description has to be added.

Comment accepted and image corrected in the paper.

Fig.6 – annotation + scale in Fig.6a is not readable.

Comment accepted and image corrected in the paper.

Fig.7 and 8 – axis description. Fig 7 and 8 show the same dependences, probably authors placed the wrong images, etc.

Comment accepted and image corrected in paper.

  1. Lines 193-194: Correction factors were applied to normalize stress-strain data. The used approach is quite a simplification. Maybe it should be better to use correction factors based on the elastic modulus comparison.

We agree with the comment made and think that this could be the approach in the found research with the plastic specimen.

The idea with the applied values ​​of the correction factor in this paper was to analyze the percentage of filling on the stress values.

  1. Fig.5b and 6b – very steep changes in the graphs showing von Mises strain vs. Section length suggest, that the chosen resolution could not be sufficient to reveal real strain distribution within a given section. Is this a limitation of the method, or this could be improved by adjusting the parameters?

The parameters can be adjusted to obtain a visually more even distribution of deformations on the surface of the specimen, without abrupt transitions, but this will not affect the shape of diagrams 6b and 7b.

The values ​​of the largest deformations are in the central zone of narrowing of the PRTS, while the values ​​are significantly lower in places farther from that zone.

Note that the results are shown in 6b or 7b at the time before the fracture of the specimen and that then the deformations are greater as well as the differences in the values ​​of deformations in the central narrowing zone PRTS (fracture site) and the rest of the PRTS surface.

  1. Fig.8 shows average strain vs. stage dependence. Representation of the average strain vs stress would be much more informative.

Comment accepted and image corrected in the paper.

Reviewer 2 Report

Detailed comments are as follows,

1)      In the part of inclusion, this work recommends that Single PRTS 90 % infill specimens be used for future research, because the experimental results of Single PRTS 90 % and 100% infill are similar. However, only break force of them are similar (showed in figure 3). The tensile curves of them differ greatly (showed in figure 8). Maybe the statement of the conclusion needs to be checked.

2)      It may be more appropriate to present the test conditions in a tabular format. It is also advisable to include a schematic diagram when introducing the test equipment.

3)      The expression of deviation in Figure 3 is unclear. A box plot may be more appropriate for displaying the diagram content. Besides, the unit of vertical coordinate is missing.

4)      In Figure 5, the units of coordinates are missing.

5)      In Figures 6-7, the texts and curves are not clear. Remading diagrams may be necessary.

Author Response

Thank you for detailed comments.

All of these comments were analyzed and the answers are shown below:

1.In the part of inclusion, this work recommends that Single PRTS 90 % infill specimens be used for future research, because the experimental results of Single PRTS 90 % and 100% infill are similar. However, only break force of them are similar (showed in figure 3). The tensile curves of them differ greatly (showed in figure 8). Maybe the statement of the conclusion needs to be checked.

Diagram 8a shows that the deformation value for Single PRTS with 90% infill is higher than Single PRTS with 100% infill on average by 0.34% for each stage, but that Single PRTS with 100% infill has a more linear Strain-Stage dependence than Single with 90 % infill.

Diagram 8a shows that the deformation value for Single PRTS with 90% infill is higher than Single PRTS with 100% infill on average by 0.17% for each stage. but that Single PRTS with 100% infill has a more linear Strain-Stage dependence than Single with 90% infill.

Did the Reviewer mean Tensile Curves on Fig. 5a and 5b?

The results for Single PRTS 90% and 100% infill are obviously different in some places or moments in greater or lesser percentage and our recommendation to choose Single PRTS 90% is to save material and time when printing in order to test PRTS from PLA material or further improvement of the procedure.

  1. It may be more appropriate to present the test conditions in a tabular format. It is also advisable to include a schematic diagram when introducing the test equipment.

      Thanks for the helpful comment. Your proposal was our original solution for presenting test conditions, but it was abandoned by looking at solutions in the literature in this area and for technical reasons.

  1. The expression of deviation in Figure 3 is unclear. A box plot may be more appropriate for displaying the diagram content. Besides, the unit of vertical coordinate is missing.

The expression of deviation in Figure 3 is corrected in the paper. Coordinate units have been added.

  1. In Figure 5, the units of coordinates are missing.

      Comment accepted and image corrected in the paper.

  1. In Figures 6-7, the texts and curves are not clear. Remaking diagrams may be necessary.

Comment accepted and image corrected in the paper.

Reviewer 3 Report

General Evaluation

The paper describes an interesting experimental challenge concerning the evaluation of hoop stress-strain state in 3D-PPRTS. The main material used for the study is PLA (polymeric material) and thus the topic of the paper is marginal (or out) to the scope of the Metals. Also, the structure and contents of the manuscript, as well as the quality of presentation need further improvement to enhance readership. More detailed comments followed.

Scientific / Technical Comments

1.       There are too many symbols and abbreviations. A list of symbols is recommended to facilitate reading.  

2.       The quality of schematics and Figures seems poor and it is recommended to be drastically improved and the size of the illustrations needs to be enlarged to increase their legibility (see for instance Figures 5-8).

3.       Fracture morphology of the tested 3D-PPRTS specimens need to be described/documented more concisely.

4.       Figure captions are quite brief. More details are suggested to be written in the text.

5.       The References are inconsistently reported in the text and the list of citations needs to be complemented with additional and more recent publications.

Grammar/Language

The language is in general sufficient. However, a final proofreading is necessary to eliminate minor spelling errors.

Author Response

Thank you for your detailed comments.

All of these comments were analyzed and the answers are shown below:

 1. There are too many symbols and abbreviations. A list of symbols is recommended to facilitate reading.

Looking at the literature, we got the impression that works in this area are written with a lot of abbreviations that are listed in the text. Also, in thinning with similar works, we tried to reduce the number of necessary abbreviations to the minimum and to be able to explain what is the essence. The only new abbreviation we have introduced is PRTS (Pipe Ring Tensile Specimen) and all the others are known and many papers have already been mentioned.

 2. The quality of schematics and Figures seems poor and it is recommended to be drastically improved and the size of the illustrations needs to be enlarged to increase their legibility (see for instance Figures 5-8).

Comment accepted and image corrected in the paper.

 3. Fracture morphology of the tested 3D-PPRTS specimens need to be described/documented more concisely.

Thanks for the great comment. Our idea, in this paper, was not to deal in detail with the morphology of fractures for several reasons, and the fact that these are 3D printed specimens is one of the most important. The above comment will certainly apply further research and analysis of steel tube fractures.

 4. Figure captions are quite brief. More details are suggested to be written in the text.

Comment accepted and image corrected in the paper.

5. The References are inconsistently reported in the text and the list of citations needs to be complemented with additional and more recent publications.

Thanks for the comment. We have re-analyzed the reference list and believe that the literature is listed throughout the text in order. Newer works are included in the list of literature and listed in the introduction, there are newer works but their content and method of obtaining results is not in accordance with our method and rules of use of equipment for measuring displacement and deformation.

Round 2

Reviewer 1 Report

The authors addressed all questions and modified images according to the comments, therefore I recommend accepting the manuscript.

Author Response

Thank you for your detailed comments and useful tips that are useful for our further work.

Reviewer 3 Report

The manuscript has been moderately improved based on authors' revisions. It seems that the number of References has not been changed from the previous version.  It is suggested to include more recent citations if they were available and relevant to the present work. 

Author Response

Thanks for the helpful comment and patience. Several significant references to the more recent publication date (in the last 2 years) have been added to the paper.
